# Non-markovian electron tunneling in SARS-CoV-2 virus infection in structured environments

**Muhammad Waqas Haseeb**[1], **Mohamad Toutounji**[2]*

**1** Department of Physics, United Arab Emirates University, Al-Ain, Abu Dhabi, United Arab Emirates,
**2** Department of Chemistry, United Arab Emirates University, Al-Ain, Abu Dhabi, United Arab Emirates

◉ These authors contributed equally to this work.
* Mtoutounji@uaeu.ac.ae

## Abstract

The role of non-Markovian quantum effects in biological processes is increasingly recognized, yet remains largely unexplored in virology. This study investigates the influence of quantum tunneling on SARS-CoV-2 infection dynamics, focusing on the interaction between the viral spike protein and the host ACE2 receptor. We employed the non-Markovian quantum state diffusion (NMQSD) to model electron transfer mediated by vibrational modes within the structured lipid membrane. Our results challenge conventional semiclassical models by demonstrating that this interaction operates not in the weak coupling limit, but in an intermediate coupling regime where quantum coherence is pivotal. We find that specific vibrational modes significantly enhance tunneling efficiency and sustain coherence over extended timescales. Furthermore, our findings reveal a critical dependence on resonance: tunneling is highly efficient below resonance via resonance-assisted mechanisms, whereas tunneling rates decrease sharply above resonance due to decoherence effects. These results, which parallel coherence phenomena in photosynthesis, suggest that the host's membrane environment actively optimizes electron transfer. This work presents a new paradigm for virus-host interactions and identifies the modulation of vibrational frequencies as a potential new avenue for antiviral drug design.

## Introduction

Spatially extended quantum systems interacting with structured environments exhibit complex, non-Markovian dynamics due to environmental memory effects. These effects challenge conventional open-system techniques, necessitating reliable simulations to understand various experimental platforms and quantum devices [1–3]. For instance, in photosynthesis, quantum coherence results from the interaction between electronic couplings and vibrational modes, and similar dynamics are observed in trapped ions and molecular junctions [4–6]. Experiments have also demonstrated the sensitivity of quantum systems like "giant" superconducting atoms and quantum

**Data availability statement:** The data underlying the results presented in the study are available from Zenodo (https://zenodo.org/records/18873662).

**Funding:** This work was funded by UAE University Research Affairs under grant number 439 G-00003550.The funder had no role in study design, data collection and analysis, 440 decision to publish, or preparation of the manuscript.

**Competing interests:** NO authors have competing interests.

registers to such memory effects [7,8], influencing phenomena like quantum error propagation and the functionality of micromechanical resonators and photonic crystals [9,10].

To accurately model these dynamics, researchers must move beyond standard Born-Markov approximations that result in time-local master equations [11]. Recent theoretical advancements have focused on developing simulation techniques that handle non-Markovian dynamics effectively. These include approximate methods that adjust the system-environment boundary to validate Born-Markov approximations and numerically exact methods that leverage specific structures of the bath Hamiltonian. These approaches are particularly effective for systems with small Hilbert spaces or specific spectral densities, though determining their applicability can be challenging.

Olfaction plays a pivotal role in how organisms perceive their environment, enabling the detection and differentiation of scents through the interaction with volatile odorant molecules [12]. Traditionally, olfaction has been explained by the lock-and-key model, where the physical and chemical characteristics of odorant molecules activate G-protein coupled receptors (GPCRs), triggering neural responses [13]. However, emerging studies challenge this model by proposing that olfaction may also involve the detection of molecular vibrations, suggesting a more intricate mechanism that includes vibrational spectra sensing and electron transfer [14–19].

The natural approach to addressing these problems involves treating the odorant molecules as a perturbation and calculating the transition rates based on the semiclassical Marcus-Jortner theory. The semiclassical Marcus-Jortner theory, which assumes $\omega_c \ll k_B T$, provides a foundational approach for calculating transition rates by treating odorant molecules as a perturbation. However, recent studies using the Master equation approach challenge its validity in capturing vibrationally-assisted electron transfer dynamics, even in low-frequency environments. The theory fails to account for strong dissipation effects and lacks the ability to resolve odorant frequency selectivity under realistic biological conditions [20–23]. Incorporating environmental dissipation and multiple vibrational modes reveals richer dynamics beyond the Marcus-Jortner framework. The Diósi-Gisin-Strunz NMQSD equation has emerged as a crucial tool for understanding the dynamics of open quantum systems, particularly those interacting with non-Markovian bosonic environments [24,25]. Traditional Lindblad Markov master equations fall short in accurately describing systems where the environment-system coupling is strong or the environment is structured [26]. The NMQSD equation has proven effective in modeling non-Markovian dynamics across various systems, from finite dimensional setups like multi-spin systems and N-level systems to continuous variable systems including quantum Brownian motion, coupled cavities, and optomechanical oscillators. More recently, the equation has been adapted to also handle systems interacting with fermionic environments by incorporating Grassmann-type noise and under biological complexes [27], broadening its applicability and utility in quantum mechanics.

Amidst the ongoing challenges posed by the COVID-19 pandemic, there is a heightened imperative to deepen our understanding of viral mechanisms, particularly

how viruses like SARS-CoV-2 interact with host cells [5,20,28]. The interaction between SARS-CoV-2's spike protein and the human ACE2 receptor, crucial for viral entry, highlights the potential for quantum-level interactions to play significant roles in viral infections [29–31]. Drawing parallels from the advanced theories in olfaction, this research proposes exploring vibration-assisted electron tunneling as a potential mechanism in COVID-19 infection processes. Investigating these quantum phenomena could open novel avenues for therapeutic and preventive strategies [ 22,32].

Our study advances the NMQSD framework to examine the role of electron tunneling in the interaction between the SARS-CoV-2 spike protein and the human ACE2 receptor. We model this interaction as a quantum system coupled to a biological environment specifically, the lipid membrane using a spin-boson model to represent the two level receptor system and its coupling to an external structured bath. This approach enables us to investigate how vibrational modes of the spike protein may assist in electron tunneling at the virus receptor interface, potentially enhancing the efficiency of viral entry. By doing so, our work not only sheds light on fundamental mechanisms of viral infection but also contributes to the broader integration of quantum physics into the life sciences. The paper is organized as follows. In Section II, we present the general derivation of the non-Markovian stochastic Schrödinger equation. Section III describes the modeling approach used to address the problem. In Section IV, we discuss our results and provide a detailed analysis of the numerical findings. Finally, in Section V, we summarize our conclusions and highlight the key outcomes of the study.

## Theory and methods

A quantum system coupled to a bosonic bath is studied, described by the Hamiltonian [24] (with $\hbar = 1$):

$$H = H_{\text{sys}} + \sum_k (g_k L b_k^\dagger + g_k^* L^\dagger b_k) + \sum_k \omega_k b_k^\dagger b_k$$

(1)

where $H_{\text{sys}}$ denotes the system Hamiltonian, $L$ the Lindblad operator, $b_k$ the bath's k-th mode annihilation operator with frequency $\omega_k$, and $g_k$ the coupling strength. The bath correlation function, central to the system's dynamics, is expressed as [25]:

$$C(t, s) = \sum_k |g_k|^2 e^{-i\omega_k(t-s)}$$

(2)

By defining $z_t^* \equiv -i \sum_k g_k^* z_k^* e^{i\omega_k t}$ as a function characterizing the time-dependent states of the bath and considering $z_k$ as Gaussian random variables, $z_t^*$ becomes a Gaussian random process with zero mean $\mathcal{M}[z_t^*] = 0$ and the correlation function $\mathcal{M}[z_t z_s^*] = C(t, s)$, where $\mathcal{M}[\cdot]$ denotes the ensemble average.

At zero temperature, the system's quantum trajectory obeys a linear, time-local Quantum State Diffusion (QSD) equation:

$$\frac{\partial}{\partial t}|\psi_{z^*}(t)\rangle = [-iH_{\text{sys}} + L z_t^* - L^\dagger \bar{O}(t, z^*)]|\psi_{z^*}(t)\rangle$$

(3)

where $\bar{O}(t, z^*) = \int_0^t C(t, s) O(t, s, z^*)\, ds$, and $O$ is an operator ansatz defined by the functional derivative

$$\frac{\delta}{\delta z_s^*}|\psi_{z^*}(t)\rangle = \hat{O}(t, s, z^*)|\psi_{z^*}(t)\rangle$$

(4)

The system's reduced density operator $\rho_s(t)$ can be calculated through ensemble averages of quantum trajectories.

$$\rho_s \equiv \text{Tr}_{\text{env}}\left[e^{-iHt}|\psi_0\rangle\langle\psi_0| \otimes \rho_{\text{env},0} e^{iHt}\right] = \mathcal{M}\left[|\psi_t(z)\rangle\langle\psi_t(z)|\right]$$

(5)

The QSD approach involves deriving the functional derivative $\hat{O}$ operator, which can be exact for simple models or perturbatively derived for more general systems.

The non-Markovian unraveling of Quantum State Diffusion (QSD) predicated on states with normalization

$$\tilde{\psi}_t(z) = \frac{\psi_t(z)}{\|\psi_t(z)\|}$$

(6)

could be achieved using the Girsanov transformation [33].

$$\frac{d}{dt}\tilde{\psi}_t = -iH_{sys}\tilde{\psi}_t + (L - \langle L \rangle_t)\tilde{\psi}_t\tilde{z}_t$$
$$- \int_0^t ds\, \alpha(t,s)\left\langle (L^\dagger - \langle L^\dagger \rangle_s)\hat{O}(t,s,\tilde{z}_t) - (L^\dagger - \langle L^\dagger \rangle_s)\hat{O}(t,s,\tilde{z}_t) \right\rangle \tilde{\psi}_t$$

(7)

Where $\tilde{z}_t$ is the shifted noise,

$$\tilde{z}_t = z_t + \int_0^t ds\, C(t,s)\langle L^\dagger \rangle_s$$

(8)

and $\langle L^\dagger \rangle_s = \langle \tilde{\psi}_t|L|\tilde{\psi}_t \rangle$ is the quantum average.

The non-linear non-Markovian QSD equation can have the compact form by introducing $\Delta t(A) = A - \langle A \rangle_t$

$$\frac{d}{dt}\tilde{\psi}_t = -iH_S\tilde{\psi}_t + \Delta_t(L)\tilde{\psi}_t z_t^* - \Delta_t(L^\dagger)\bar{O}(t,\tilde{z})\tilde{\psi}_t + \Delta_t(L^\dagger)\bar{O}(t,\tilde{z})_t\tilde{\psi}_t$$

(9)

Where

$$\bar{O}(t,z^*) = \int_0^t ds\, C(t,s)\hat{O}(t,s,z)$$

(10)

The (Eq. 7 or 9) is fundamental for NMQSD and the perturbative treatment starts with this equation. Applying the formal Perturbation theory on operator $\hat{O}(t,s,z)$ using a series expansion in powers of $(t-s)$ [25].

$$\frac{d}{dt}\tilde{\psi}_t = -iH_{Sys}\tilde{\psi}_t + \Delta_t(L)\tilde{z}_t - g_0(t)((\Delta_t(L^\dagger)L - \langle\Delta_t(L^\dagger)L\rangle_t))\tilde{\psi}_t + ig_1(t)(\Delta_t(L^\dagger)[H,L]$$
$$- \langle\Delta_t(L^\dagger)[H,L]\rangle_t))\tilde{\psi}_t + g_2(t)((\Delta_t(L^\dagger)[L^\dagger,L]L - \langle\Delta_t(L^\dagger)[L^\dagger,L]\rangle_t))\tilde{\psi}_t$$

(11)

In the context of NMQSD (QSD), the first-order corrections to the zeroth-order term are governed by the characteristic system frequency $\omega$ and the environmental relaxation rate $\Gamma$. These corrections become significant when the environmental correlation time $\tau$ is finite but still shorter than the intrinsic timescales of the system, thereby justifying the use of the perturbative expansion of the QSD equation. As $\tau$ approaches zero, the memory effects of the environment diminish, and the dynamics converge to the Markovian limit, wherein only the zeroth-order term contributes. This transition delineates the boundary between Markovian and non-Markovian behavior, emphasizing the critical role of environmental memory in determining the nature of the system's quantum dynamics.

The bath can be completely characterized by its spectral density $J(\omega) = \sum_k |g_k|^2 \delta(\omega - \omega_k)$, from which we can calculate its autocorrelation function in thermal equilibrium at temperature $T$:

$$C(t) = \int_0^\infty d\omega J(\omega)[\coth\left(\frac{\omega}{2T}\right)(\cos(\omega t) - i\sin(\omega t))].$$

(12)

If this function decays slowly compared to the system time scales then the bath memory is important.

We consider a bath in which a narrow band of modes dominate the interaction with the system. The specific form of spectral density we use is [34]

$$J(\omega) = \sum_i (J_i(\omega))$$

(13)

where

$$J_i(\omega) = \frac{\gamma\Gamma\omega_i^2\omega}{(\omega_i^2 - \omega^2)^2 + \Gamma^2\omega^2}$$

(14)

Here, $\gamma$ gives the coupling strength, $\omega_i$ are the frequencies characterizing the dominant modes of the virus spike protein., and $\Gamma$ provides a measure of the width of the dominant band. The overall effect of the environment can be captured by the reorginazational energy.

$$\lambda = \int_0^\infty \frac{J(\omega)}{\omega} d\omega$$

(15)

The structured spectral density plotted in Fig 1 is represented by the coupling of the dimer system not only to its background continuum environment but also to a specific lossy mode of vibration. This dual interaction framework allows us to more accurately reflect the complex dynamics often observed in quantum systems where discrete vibrational modes significantly influence the overall system behavior. Where $\gamma^{-1}$ is the environmental memory time which correspond to the

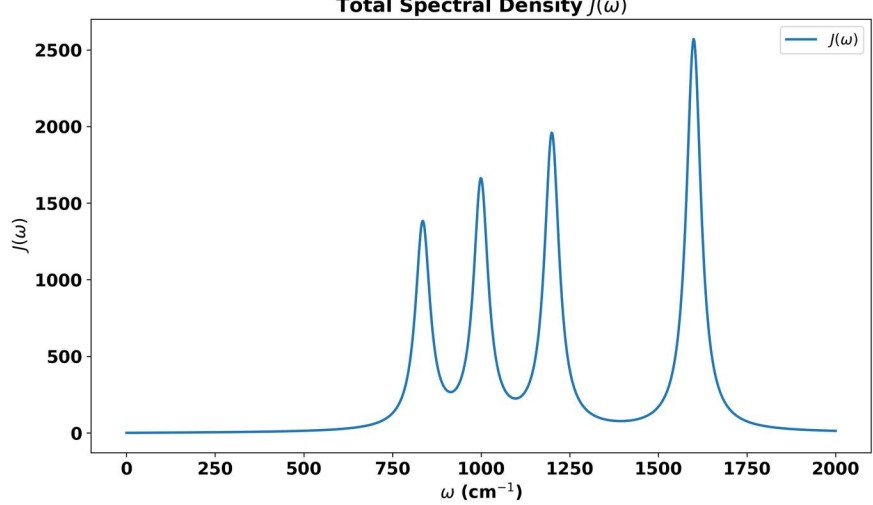

**Fig 1. The spectral density, as detailed in (Eq. 13), characterizes the system by incorporating narrow vibrational modes** $\omega_i$ **= 836, 1040, 1240, 1600cm$^{-1}$, of the virus.** These modes play a crucial role in the dynamic interactions within the system, providing a detailed insight into the mechanisms that govern the behavior of the virus at a molecular level.

cutoff frequency $\omega_c$. The combined influence of four vibrational modes within the spectral density is analyzed and represented through the autocorrelation function. The Fig 2 illustrates the interaction of vibrational modes, offering insights into the system's dynamic response to environmental coupling. It highlights how increasing environmental coupling significantly enhances the correlation function. The Marcus-Jortner regime, limited to very weak coupling, fails to capture the true dynamics of viral infection mechanisms. By extending to the intermediate coupling regime, where non-Markovian effects emerge, we better reflect the dynamics of viral infection under realistic biological conditions.

The Quantum State Diffusion (QSD) equation plays a fundamental role in modeling the dynamics of open quantum systems, particularly in capturing the influence of environmental interactions. It incorporates the system Hamiltonian, the Lindblad operator, and their commutators to describe stochastic evolution under environmental noise. A central aspect of applying the QSD formalism involves determining the time-dependent coefficients $g_i(t)$, which are derived from the environmental correlation function $C(t,s)$. These coefficients encapsulate the statistical structure of quantum noise and are especially informative when the environment is characterized by Ornstein-Uhlenbeck processes, which feature exponentially decaying correlations. Accurate evaluation of $g_i(t)$ is essential for utilizing the QSD framework in realistic scenarios where quantum systems are exposed to structured or noisy environments.

$$g_0(t) = \int_0^t C(t, s)ds \tag{16}$$

$$g_1(t) = \int_0^t C(t, s)(t - s)ds \tag{17}$$

$$g_2(t) = \int_0^t du \int_0^s C(t, s)C(s, u)(t - s)ds \tag{18}$$

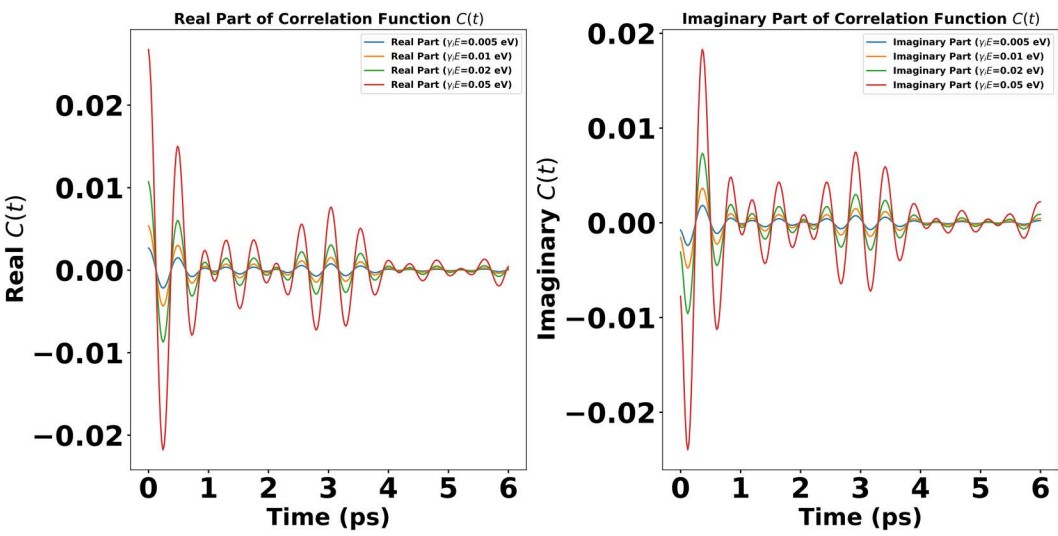

**Fig 2. Autocorrelation Function in Thermal Equilibrium when $\Gamma$ = 0.005$eV$, $\omega_i$ = 836, 1040, 1240, 1600$cm^{-1}$, $\gamma_{iE}$ = 0.005, 0.01, 0.02.0.05$eV$ at Temperature = 300 K. (a)** Depicts the real component of the autocorrelation function, representing the combine effect of all the vibrational modes w.r.t different coupling strengths to environment. **(b)** Showcases the imaginary component of the autocorrelation function dynamics in thermal equilibrium.

## Theoretical model

Biological processes are increasingly investigated within the framework of open quantum systems, which offers a powerful formalism for analyzing the complex interactions between a system and its surrounding environment [4]. In particular, electron transfer phenomena often modulated by vibrational dynamics are commonly described using the well established spin-boson model [23]. This model conceptualizes the system as a donor-acceptor pair coupled to an environmental reservoir, typically represented as a continuum of harmonic oscillators [11], thereby capturing both dissipative and coherent aspects of quantum transport in biological settings. In our study, we consider the complex of a ligand protein and its receptor, along with their immediate surroundings, as a cohesive system. Drawing inspiration from theories proposed for olfaction [35], we utilize open quantum systems to probe how interactions between the ACE2 receptor and a vibrational mode of the SARS-CoV-2 spike protein influence electron transfer probabilities. The receptor is modeled as a dimer, with the principle Hamiltonian detailing the dynamics within this system and its interactions with the surrounding environment. The Hamiltonian for the receptor is

$$H_R = \frac{1}{2}\epsilon\sigma_z + \frac{1}{2}\Delta\sigma_x \tag{19}$$

For a dimer isolated from external interaction, the maximum probability of a transition from donor to acceptor is given by [36,37]

$$\text{Max}[P_{D\to A}(t)] = \frac{\Delta^2}{\Delta^2 + \epsilon^2} \tag{20}$$

where $\epsilon = \epsilon_D - \epsilon_A$. When the energy of the donor and the acceptor are equal, i.e., $\epsilon_D = \epsilon_A$, the maximum transfer probability from donor to acceptor [$P_{D\to A}(t)$] at time $t_0 = \pi/2\Delta$ is equal to 1. The Hamiltonian of the ligand, in this case the spike protein, is represented as a harmonic oscillator with frequencies $\omega$ associated with the protein:

$$H_P = \sum_i \omega_i(b^\dagger b + 1/2) \tag{21}$$

$$H_{R-P} = \sigma_z \sum_i \omega_i\gamma_i(b + b^\dagger) \tag{22}$$

The interaction Hamiltonian of our model incorporates both the donor and acceptor, as well as the ligand protein, though interactions involving the ligand are reduced for computational efficiency. The strength of the interaction between the ligand protein and its receptor is quantified by the coupling constant $\gamma$. Vibrational states of the SARS-CoV-2 spike protein are modeled using creation and annihilation operators, $b^\dagger$ and $b$, corresponding to the vibrational mode $\omega$. Our system is detailed through Hamiltonians $H_R$, $H_P$, and $H_{R-P}$, which focus on the dynamics of electron tunneling critical to the receptor interactions, facilitating a nuanced exploration of the quantum dynamics influencing viral entry mechanisms.

$$H_{sys} = \frac{1}{2}\epsilon\sigma_z + \frac{1}{2}\Delta\sigma_x + \sum_i \omega_i(b^\dagger b + 1/2) + \sigma_z \sum_i \omega_i\gamma_i(b + b^\dagger) \tag{23}$$

Similarly, the membrane environment is characterized by $H_E$, and its interplay with the receptor is approximated as $H_{R-E}$.

$$H_E = \sum_E \omega_E(b_E^\dagger b_E + \frac{1}{2}) \tag{24}$$

$$H_{R-E} = \sigma_z \sum_{i,E} \omega_E \gamma_{iE}(a_E + a_E^\dagger)$$

(25)

In the equation, the term $\gamma_{iE}$ denotes the interaction strength between the receptor and its surrounding membrane environment, which is considerably weaker than its interaction with the spike protein. The Hamiltonians $H_{R-P}$ and $H_{R-E}$ specify different interaction dynamics: $H_{R-P}$ facilitates the receptor's recognition of the spike protein, while $H_{R-E}$ accounts for the receptor's general coupling with environmental vibrational modes. This framework allows for a clear differentiation between the vibrational modes of the spike protein and those of the surrounding environment by leveraging their distinct frequency profiles and the relative strengths of their coupling constants. The summation terms in the interaction Hamiltonians collectively represent the contributions from the donor, acceptor, and all relevant environmental vibrational modes, thereby capturing the intricate interplay of coherent and dissipative dynamics within the system.

In our analysis, the system consists of the SARS-CoV-2 spike protein and the ACE2 receptor. The Hamiltonian describing this system is composed of $H_R$, $H_P$, and $H_{R-P}$. Additionally, the Lindblad operator for this configuration is specified as $L = \gamma_{iE}\sigma_z$. As a result, in the non-Markovian regime, our equation is outlined in (Eq. 11), capturing the complex dynamics between the spike protein and the receptor.

$$\frac{d}{dt}\tilde{\psi}_t = -iH_{sys}\tilde{\psi}_t + \gamma_{iE}\left(\sigma_z - \langle\sigma_z\rangle_t\right)\tilde{\psi}_t\tilde{z}_t + g_0(t)\gamma_{iE}^2\left(\langle\sigma_z\rangle_t\sigma_z - \langle\sigma_z\rangle_t^2\right)\tilde{\psi}_t$$
$$+ g_1(t)\gamma_{iE}^2\Delta\left[\left(\sigma_z - \langle\sigma_z\rangle_t\right)\sigma_y - \langle\left(\sigma_z - \langle\sigma_z\rangle_t\right)\sigma_y\rangle_t\right]\tilde{\psi}_t$$

(26)

where

$$\tilde{z}_t = z_t + \gamma_{iE}\int_0^t ds\, C(t-s)\langle\sigma_z^\dagger\rangle s$$

(27)

The coefficients $g_0(t), g_1(t)$ are given by (Eq. 12) and (Eq. 13), respectively.

By numerically simulating the dynamics governed by Eq. 26 and computing the ensemble-averaged density matrix as defined in Eq. 5, we extract the maximum probability of electron transfer within the dimer system in the presence of vibrational modes. To quantify the contribution of these vibrational modes, we evaluate the difference in transfer probabilities between systems with and without vibrational coupling. This quantity is expressed as follows [36]:

$$\Delta P = Max[P_{D\to A}(t)]_{with} - Max[P_{D\to A}(t)]_{without}$$

(28)

The observable $\Delta P$ was chosen to highlight the influence of vibrational modes on the maximum population transfer. This choice was motivated by our focus on understanding how vibrational coupling affects the peak efficiency of electron transfer. Since vibrationally assisted tunneling can enhance specific time-dependent transitions, $\Delta P$ serves as a direct measure of this enhancement. Furthermore, the rates with and without the virus can be calculated from the populations using the density matrix as follows:

$$\kappa = -\frac{d}{dt}\ln\left(P(t)\right)$$

(29)

## Numerical results and discussion

We quantify the maximum probability of electron transfer between donor and acceptor states as a function of the dimer's detuning and coupling parameters. By solving the non-Markovian Stochastic Schrödinger equation (Eq. 26) within a

structured environment and computing the ensemble-averaged density matrix, we evaluate how these parameters influence tunneling dynamics. We Solve (Eq. 26) with an Euler–Maruyama (Itô) scheme. For the state $\psi(t)$ we solve with Lindblad operator $L = \gamma_E \sigma_z$, Gaussian Wiener increment $dW_t \sim \mathcal{N}(0, dt)$, and Hermitian $H$ as defined in Eqs. (1)–(11). Unless otherwise noted we use a fixed time step $\Delta t = 0.005$ and total propagation time $T = 4$ ps. Observables are computed as ensemble means over $N = 1000 - 2000$ independent trajectories per condition; in two-condition comparisons we employ a paired design (the same random seed per trajectory across conditions) to reduce variance. Uncertainty is reported as mean ± 95% confidence intervals obtained by nonparametric bootstrap over trajectories (typically $B = 3000$ for time series and $B = 5000$ for parameter-sweep heat maps). Numerical convergence is verified by halving the time step ($\Delta t/2$) and reporting the relative change in terminal donor survival population; in all figures this change is < 2% (the exact value is listed in each caption). Random numbers are generated with a fixed seed range $\{s_0, \ldots, s_0 + N - 1\}$ for full reproducibility, and the simulation scripts output the exact $\Delta t$, $T$, $N$, seeds, and Hamiltonian parameters used. All statistical analyses are presented in the Supporting Information (S1).

To visualize the effect of vibrational coupling, we construct a two-dimensional histogram that compares the maximum electron transfer probabilities in the presence and absence of vibrational modes. This comparison is captured by the observable $\Delta P$, defined as $\Delta P = \mathrm{Max}[P_{D \to A}(t)]_{\text{with vibrational mode}} - \mathrm{Max}[P_{D \to A}(t)]_{\text{without vibrational mode}}$, which serves as a direct measure of the enhancement attributable to vibrationally assisted tunneling.

We have collected vital parameters from diverse biological processes to refine our analysis, specifically integrating models from Solov'yov et al [35,38] that examine vibration-assisted electron tunneling in olfactory receptors. We are considering the only the peaks of the vibrational modes present in Raman Spectroscopy experiment [38] and assume that without the peaks modes only contribute to support particular frequency selection. These parameters, essential for the accuracy of our model, are elaborately detailed in Table 1. All energies are reported in eV; conversions use 1 eV = 8065.54 cm$^{-1}$ (i.e., 1 cm$^{-1}$ = 1.23984 × 10$^{-4}$ eV), and times use 1 ps = 10$^3$ fs.

Our study examines how vibrational frequencies influence electron transfer dynamics between the SARS-CoV-2 spike protein and its receptor across a range of coupling strengths, with dimer couplings set at $\Delta = 0.0001, 0.001, 0.01, 0.1$ eV [23]. To inform the selection of vibrational modes, we also incorporate insights from Raman spectroscopy analyses of SARS-CoV-2 [38]. At the lowest coupling strength, $\Delta = 0.0001$ eV, shown in Fig 3(a), the vibrational modes exhibit negligible influence on electron transfer probability. This suggests that weak donor–acceptor coupling is insufficient to facilitate significant electron tunneling, even in the presence of strong vibrational interactions. As the coupling increases to $\Delta = 0.001$ eV, illustrated in Fig 3(b), a modest enhancement in transfer probability is observed, indicating the initial emergence of vibrationally assisted tunneling effects.

Increasing the coupling strength to $\Delta = 0.01$ as shown in Fig 3(c) leads to a notable enhancement in electron tunneling and transfer probabilities, underscoring the increasing importance of vibration-assisted electron tunneling mechanisms. At the maximum coupling strength evaluated, $\Delta = 0.1$, illustrated in Fig 3(d), the vibrational mode plays a crucial role in significantly boosting the transfer probability, demonstrating how strong coupling combined with vibronic assistance can significantly improve electron transfer. This finding highlights the essential role of coupling strength in influencing the interactions between the spike protein and the ACE2 receptor, thereby impacting the virus infection dynamics at the molecular

**Table 1. Parameters used in the numerical simulations.**

| Parameter | $\epsilon_A - \epsilon_D$ | $\Delta$ | $\gamma_i$ | $\gamma_{iE}$ | $\omega_1$ | $\omega_2$ | $\omega_3$ | $\omega_4$ |
|---|---|---|---|---|---|---|---|---|
| Range | 500–1700 cm$^{-1}$ | 0.0001–0.1 eV | 0–0.49 eV | 0.005–0.025 eV | 836 cm$^{-1}$ | 1000 cm$^{-1}$ | 1240 cm$^{-1}$ | 1600 cm$^{-1}$ |

The parameters for the numerical simulations are chosen following Refs. [35,38].

**Fig 3. This figure illustrates the impact of the vibrational mode of the SARS-CoV-2 spike protein on electron transfer probabilities across a range of dimer coupling strengths, as shown in panels (a)–(d).** In panel **(a)**, corresponding to a weak coupling regime with $\Delta = 0.0001$ eV, the vibrational mode has minimal influence on the electron transfer probability. Panel **(b)** shows a slightly increased coupling strength at $\Delta = 0.001$ eV, where a modest enhancement in transfer probability is observed. In panel **(c)**, with $\Delta = 0.01$ eV, the effect of the vibrational mode becomes more pronounced, leading to a noticeable increase in transfer efficiency. Finally, panel **(d)** represents the strong coupling regime at $\Delta = 0.1$ eV, where the vibrational mode significantly enhances electron transfer, as indicated by the prominent white region in the plot.

level. In our recent publication [27], we reported analogous results in a structureless environment. However, when analyzing the tunneling probability within a structured environment, we observed a more pronounced effect compared to that in a Lorentz-Drude environment. This underscores the significant impact of environmental structure with narrow modes on electron tunneling dynamics, highlighting the necessity to consider the specific environmental characteristics when evaluating quantum mechanical interactions and processes. Such insights are vital for developing targeted strategies to inhibit viral entry and infection. Moreover, the mechanism of electron transfer within these molecular systems is critical for recognizing molecular signatures, particularly through detecting vibrational spectra linked with the spike protein, as described in [23]. This sensitivity to vibrational signatures enhances our understanding of the fundamental electron transfer dynamics and their role in molecular interactions, contrasting sharply with prior studies in the Markovian regime where very strong couplings led to non-physical negative probabilities [27,39].

Fig 4 illustrates the comparison of decay rates ($\kappa$) as a function of time ($t$) for two scenarios: with the virus (blue curve) and without the virus (red curve). When the parameter values are set as $\Delta = \gamma_{iE}$ and $\omega = 1000$ cm$^{-1}$(0.1040eV), the decay rate exhibits significant oscillatory behavior in the presence of the virus. These oscillations, a hallmark of non-Markovian dynamics, reflect the memory effects influencing the system's evolution. Non-Markovianity implies that the system retains coherence or information about its past interactions, resulting in feedback between the system and its environment. Over

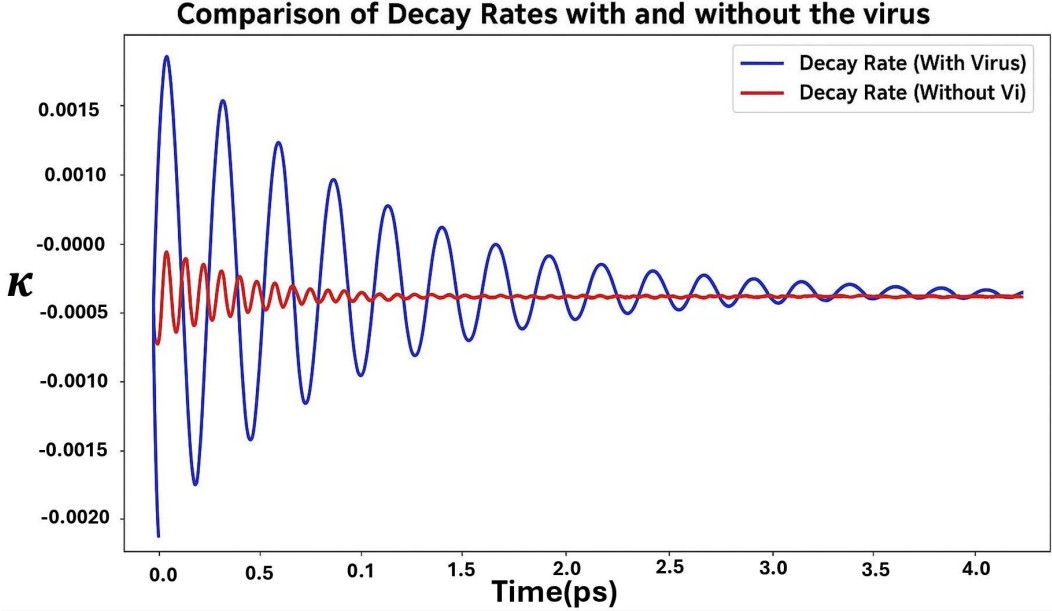

**Fig 4. Comparison of Decay Rates ($\kappa$) with and without the Virus for $\Delta = \gamma_{iE}$ and $\omega$ = 1000 cm⁻¹.** The blue curve, representing the decay rate with the virus, demonstrates non-Markovian dynamics with pronounced oscillations due to memory effects, while the red curve, representing the decay rate without the virus, shows dynamics with nearly constant behavior.

time, these oscillations gradually decay, indicating that the strong correlations induced by the interaction with the virus diminish, allowing the system to transition to a steady state. However the decay rate without the virus remains nearly constant.

Our detailed investigation spans the examination of population and coherence dynamics across a spectrum of vibronic modes at $\omega_i$ = 836, 1000, 1240, 1600cm⁻¹(0.1040–0.1886 eV), extending from weak to strong coupling domains between the donor and acceptor levels and from weak to moderate coupling between the environment. We begin our analysis by considering the highest peak in the Raman spectra, which corresponds to $\omega$ = 1000 cm⁻¹. Figs 5, 6(a-d) display these dynamics. Particularly, Fig 5(a) shows the population dynamics at the weakest coupling $\Delta$ = 0.0001 eV, where populations are minimal, mirroring earlier observations in Figs 3(a-d), suggested limited electron transfer under similar conditions.

Increasing the coupling to $\Delta$ = 0.001eV results in a slight enhancement in transfer probabilities, suggesting increased electron mobility. This trend continues as seen in Fig 5(c) at $\Delta$ = 0.01 eV, where the impact of vibronic modes becomes more pronounced, particularly in with the virus, which exhibits superior populations. The culmination of these trends is observed at the highest coupling strength evaluated, $\Delta$ = 0.1 eV, as shown in Fig 5(d). At this juncture, the structured environment not only enhances the transfer probabilities but also dramatically amplifies the population dynamics, thereby facilitating a more robust interplay between the quantum states. Furthermore, Fig 5(d) illustrates the coherent regime, characterized by $\Delta \ggg \gamma_{iE}$, where the transfer occurs due to the strong coupling between the donor and acceptor. These observations align with our previous work, which demonstrated that electron tunneling predominantly occurs in the intermediate ($\Delta$ = 0.01 eV) to strong ($\Delta$ = 0.1 eV) coupling regime.

Fig 6 Coherence across $\Delta$ with and without the Spike environment. Across the $\Delta$ sweep, we compare dynamics without the Spike to dynamics with the Spike using the non-Markovian stochastic Schrödinger framework. At very weak coupling 6 (a) ($\Delta$=0.0001eV), coherence is essentially absent without the Spike—populations remain near their initial values as dephasing and localization dominate—whereas the with Spike environment sustains small, underdamped oscillations and visibly higher coherence because its oscillatory memory feeds phase back into the two-level system. Increasing to 6 (b)

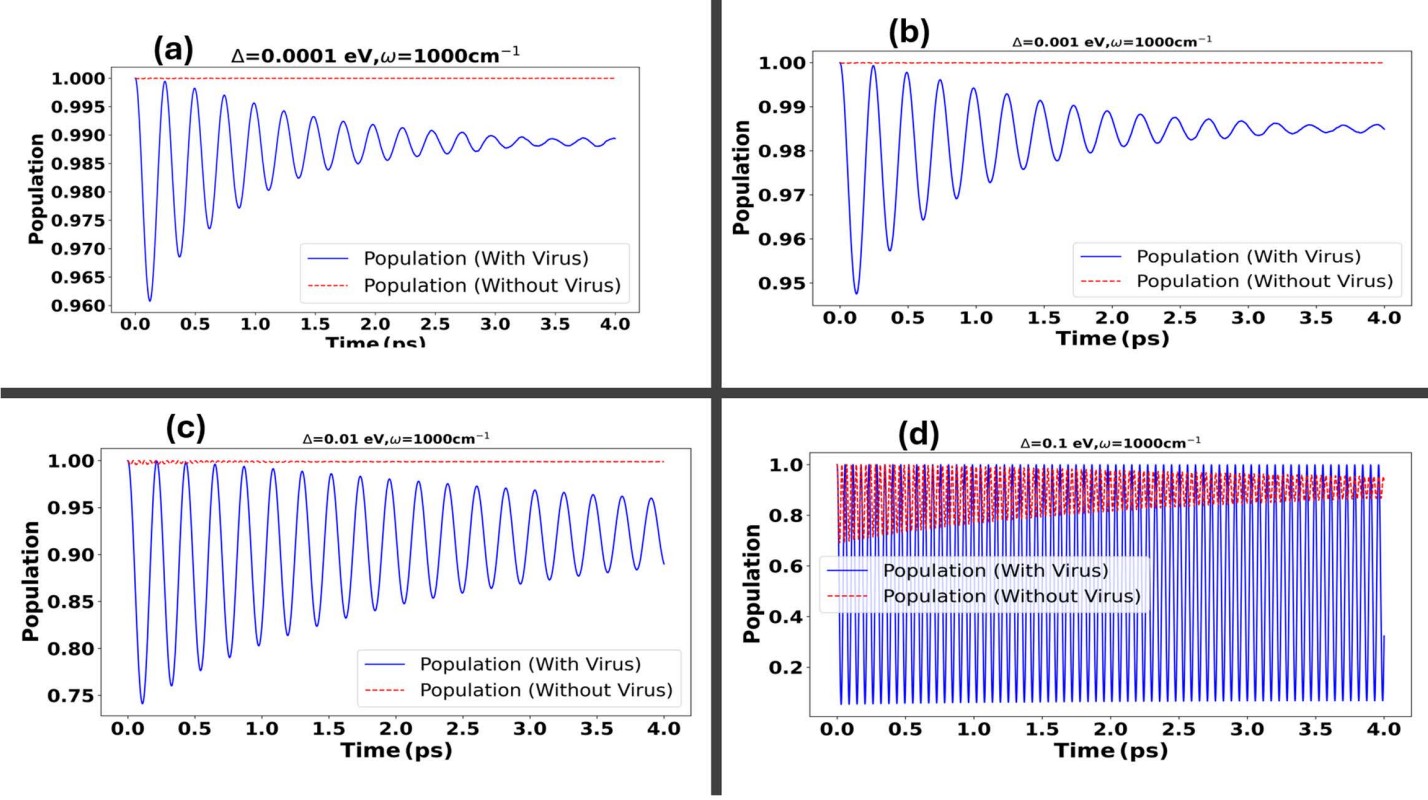

**Fig 5. Comparison of Population Dynamics in Structured Environments Across Various Coupling Strengths with (blue) and without (red) the Virus Spike protein.:** $\omega = 1000\,\mathrm{cm}^{-1}$, $\gamma_{iE} = 0.01\,eV$, $\lambda = 0.015\,eV$, $\Delta = 0.0001 - 0.1\,eV$. **(a)** At the weakest coupling of $\Delta = 0.0001\,eV$, both cases exhibit very low transfer probabilities, indicating minimal electron transfer activity. **(b)** Increasing the coupling strength to $\Delta = 0.001\,eV$ results in a modest enhancement in transfer probabilities across with the virus, with the vibrational mode beginning to exert a noticeable influence. **(c)** At an intermediate coupling strength of $\Delta = 0.01\,eV$, the impact of the vibrational mode becomes significantly more pronounced with the virus in the structured environment, leading to distinctly higher transfer probabilities compared to without the spike. **(d)** In the strong coupling regime at $\Delta = 0.1\,eV$, the vibrational mode dramatically enhances electron transfer in the structured environment, markedly surpassing the dynamics observed without the virus.

$\Delta$=0.001eV produces no qualitative change in the reference case; with the Spike, the coherence envelope remains clearly discernible before relaxing as dissipative feedback accumulates. At intermediate coupling 6 (c) ($\Delta$=0.01eV), coherence grows in both conditions, but it is more pronounced with the Spike: the structured correlations enhance the amplitude and lifetime of early-time beats before damping takes over. At strong coupling 6 (d) ($\Delta$=0.1eV), fast, high-contrast oscillations appear in both scenarios, and the Spike's vibrational interaction further shapes phase and decay through beating, showing how a resonant, memory-bearing environment can sustain and steer quantum coherence more effectively than a featureless bath.

The structure and influence of the environment play a critical role in explaining the dynamics of biological complexes. In this analysis, we advance our study by fixing the tunneling term $\Delta$ and examining the effect of the structured environment, transitioning from the weak to moderate coupling regime, on electron transfer dynamics. Figs 7, 8(a-d) display of population and coherence dynamics across all of vibrational modes at $\omega_i = 836, 1000, 1200, 1600\,\mathrm{cm}^{-1}$, exploring the transition from weak to moderate coupling with the environment. Figs 7(a) illustrate the population dynamics for $\Delta = 10\gamma_{iE}$ and $\lambda = 0.007\,eV$, where the influence of the environment is minimal, and the dynamics are predominantly governed by the vibrational mode. A pronounced effect is observed as the vibrational frequency transitions from $\omega_i = 836 - 1200\,\mathrm{cm}^{-1}$,

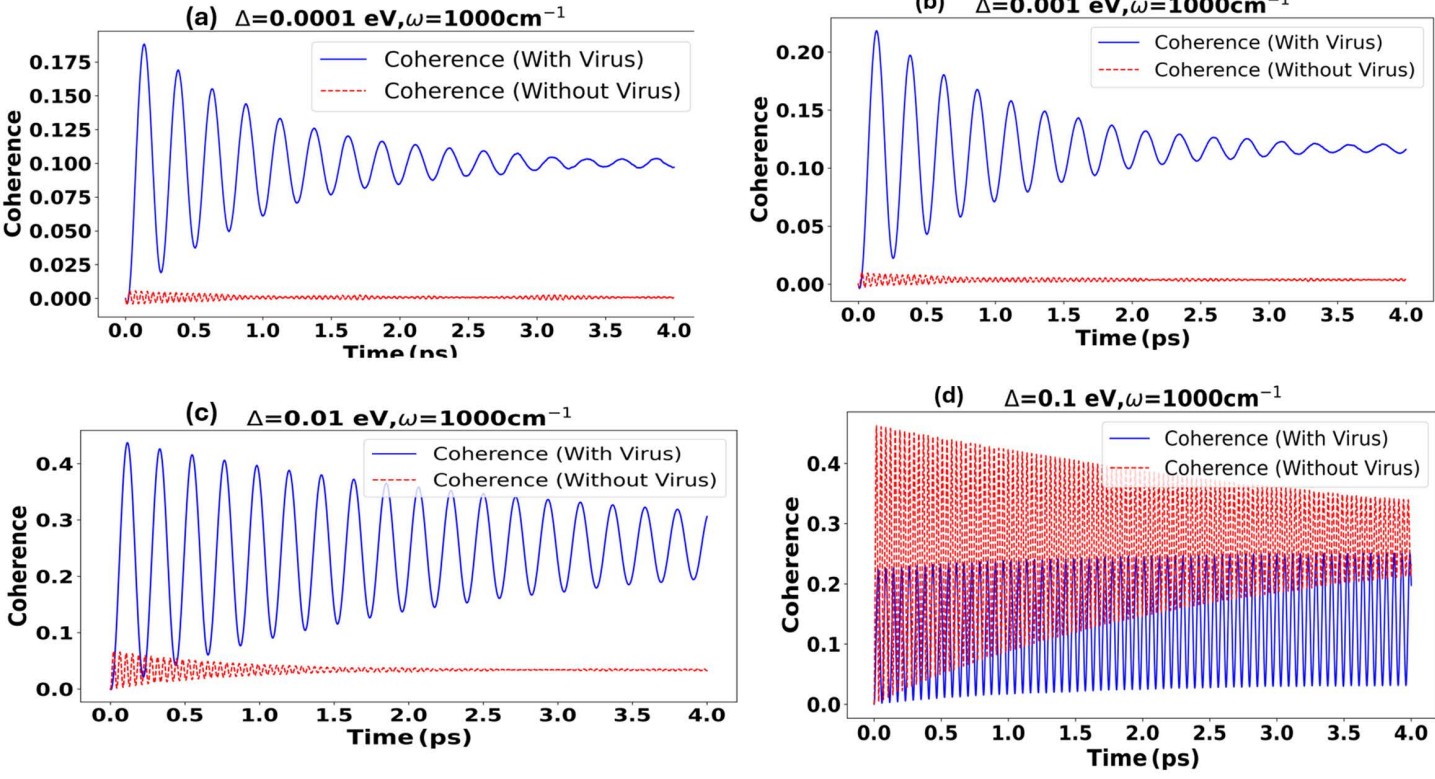

**Fig 6. Coherence dynamics were analyzed using the non-Markovian Stochastic Schrödinger Equation across different coupling strengths and a vibrational mode of 1000 cm⁻¹, presented in panels (a-d) with (blue) and without(red) under Structured Environments. (a)** At a weak coupling of $\Delta = 0.001eV$, the coherence is minimal without the virus, whereas with the virus maintains higher coherence even at minimal coupling. **(b)** Increasing the coupling to $\Delta = 0.001eV$ reveals no marked change in coherences. **(c)** An intermediate coupling of $\Delta = 0.01eV$ enhances coherence, with the virus displaying more pronounced effects. **(d)** A strong coupling of $\Delta = 0.1eV$ markedly influences coherence dynamics, underscoring the significant impact of vibrational interactions and robust coupling on the system's quantum coherence in both cases.

attributed to resonance effects. Notably, the probability of transfer is weakest at $\omega = 1600\,\text{cm}^{-1}$, highlighting the diminishing role of vibrational coupling in this regime. As we transition from the weak coupling regime to a regime where the reorganization energy $\lambda = (0.015 - 0.03)\,eV$ approaches or exceeds the thermal energy $k_B T$ Fig 7 (b-c), in this regime the Marcus-Jortner theory fails to accurately predict the dynamics. However, our NMQSD equations reveal that, while oscillations become damped, the populations remain relatively intact within this coupling limit. When the environmental coupling exceeds the thermal energy, the environmental effects dominate visible in Fig 7(d), leading to a reduction in tunneling efficiency. Coherence dynamics for electron transfer under a structured environment influenced by the virus spike protein show a clear transition from weak to moderate coupling regimes. Figs 8 (a-d) illustrate the dynamics across vibrational modes at $\omega_i = 836, 1000, 1240, 1600\,\text{cm}^{-1}$, with a fixed tunneling term $\Delta$. In (a), when $\Delta = 10\gamma_{iE}$ and $\lambda = 0.007\,\text{eV}$, the environmental influence is minimal, allowing the dynamics to be primarily driven by the vibrational modes. Positive coherence emerges for resonant vibrational frequencies, as seen when $\omega$ transitions from $836\,\text{cm}^{-1}$ to $1000\,\text{cm}^{-1}$. However, at off-resonant vibrational modes, such as $1200, 1600\,\text{cm}^{-1}$, negative coherence dominates, indicating a mismatch between vibrational frequencies and resonance effet. In Fig 8 (b-c), as the reorganization energy increases to $\lambda = (0.015 - 0.03)\,eV$, the system enters a regime where oscillations are damped but coherence persists, reflecting the intermediate coupling effects. Finally, in (d), when the environmental coupling strength surpasses $k_B T$, strong environmental effects dominate

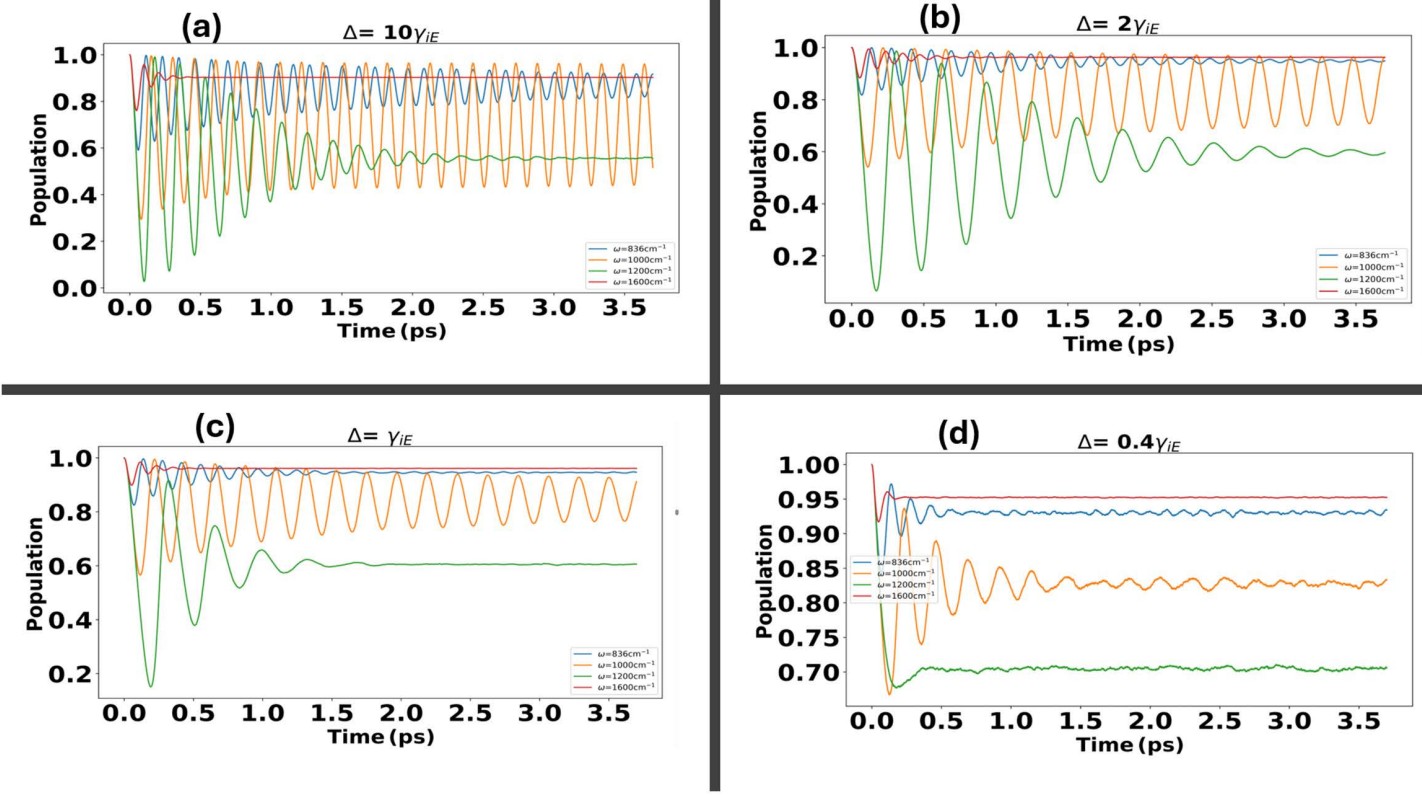

**Fig 7. Population dynamics for electron transfer under the influence of a structured environment with Virus Spike Protein, transitioning from weak to moderate coupling regimes. (a-d)** depict the dynamics across vibrational modes at $\omega_i$ = 836, 1000, 1240, 1600 cm$^{-1}$ with a fixed tunneling term $\Delta$. In **(a)**, when $\Delta = 10\gamma_{iE}$ and $\lambda = 0.007$ eV, the environment has minimal influence, with dynamics predominantly governed by the vibrational modes. Resonance effects enhance transfer probabilities as $\omega$ transitions from 836 cm$^{-1}$ to 1240 cm$^{-1}$, while the transfer weakens at 1600 cm$^{-1}$. **(b-c)** showing the transition to a regime where the reorganization energy $\lambda = (0.015 - 0.03)$ eV showing the damped oscillations with populations remaining intact. In **(d)**, moderately strong environmental coupling beyond $k_B T$, the dynamics are dominated by environmental effects, leading to a pronounced reduction in tunneling.

the dynamics, significantly suppressing coherence and reducing tunneling rates. This highlights the complex interplay between vibrational modes, tunneling, and environmental effects in determining the efficiency of electron transfer. Olfactory receptors function as vibrational spectrometers, capable of resolving distinct molecular vibrational frequencies. To analyze how these receptors discriminate specific vibrational frequencies under the influence of environmental effects, we evaluate the ratio.

$$r = \frac{\kappa - \kappa_0}{\kappa + \kappa_0}$$

(30)

This ratio quantifies the comparison between total tunneling rates with virus ($\kappa$) and tunneling rates without the virus ($\kappa_0$) as a function of the virus vibrational frequency $\omega_i$ and the bath reorganization energy $\lambda$.

Fig 9 provides a detailed contour plot that captures the interplay between environmental structure, reorganization energy, and vibrational frequencies, illustrating their combined influence on decay rates. In regions where the reorganization energy is weak, the decay rates are higher, as indicated by the red region in the plot, suggesting that the system is weakly influenced by the environment in this regime. This corresponds to a faster transfer process, driven by the

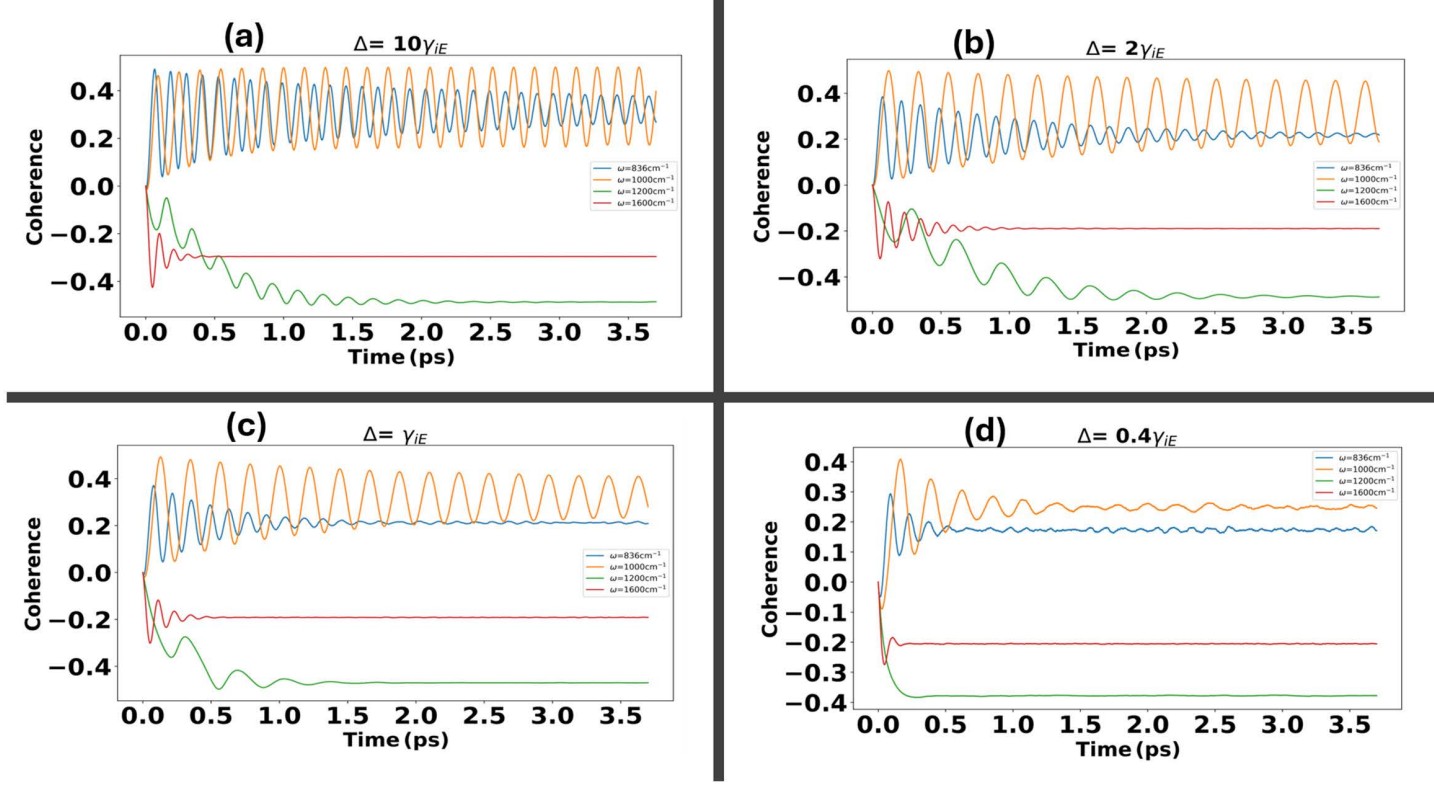

**Fig 8. Coherence dynamics for electron transfer under the influence of a structured environment with virus spike Protein, transitioning from weak to moderate coupling regimes. (a-d)** depict the dynamics across vibrational modes at $\omega_i$ = 836, 1000, 1240, 1600 cm$^{-1}$ with a fixed tunneling term $\Delta$. In **(a)**, when $\Delta = 10\gamma_{iE}$ and $\lambda$ = 0.007 eV, the environment has minimal influence, with dynamics predominantly governed by the vibrational modes. Resonance effects enhance transfer probabilities as $\omega$ transitions from 836 cm$^{-1}$ to 100 cm$^{-1}$ showing positive coherences, while the negative coherences is being seen when the modes are off resonances at 1200 cm$^{-1}$ **(a-d)**. **(b-c)** showing the transition to a regime where the reorganization energy $\lambda$ = (0.015 − 0.03) eV showing the damped oscillations. In **(d)**, moderately strong environmental coupling beyond $k_BT$, the dynamics are dominated by environmental effects, leading to a pronounced reduction in coherences.

Vibrational modes of the Virus dense and narrow spaced line in red region. As the reorganization energy approaches values comparable to the thermal energy, the system transitions to a different regime, marked by the yellow region. In this domain, the decay rates remain relatively stable despite the increase in reorganization energy. The plot also shows a denser clustering of contour lines, indicating more nuanced variations in decay rates, possibly due to thermal energy competing with the reorganization energy in driving the system dynamics.

When the reorganization energy exceeds the thermal energy, a distinct shift occurs. This results in a significant decrease in the decay rate ratio, as shown in the blue region of the figure. The increased spacing between contour lines in this region reflects a slowing of the system dynamics and a reduced sensitivity of decay rates to further increases in reorganization energy. This regime highlights the dominance of stronger coupling and vibrational mode effects, which reduce the efficiency of transfer as the system moves into a more structured and energetically distinct environment. Furthermore, Vibrational modes begin to play less and less pronounced role, when increasing the vibrational frequency as compared to energy difference between the donor and acceptor. Overall, the contour plot in Fig 9 provides a view of how reorganization energy, thermal energy, and vibrational modes collectively govern the decay rates and transfer dynamics. Moreover, the increase in vibrionic mode frequency is found to introduce more pronounced oscillatory behaviors in both the population and coherence dynamics, illustrating a dynamic and complex

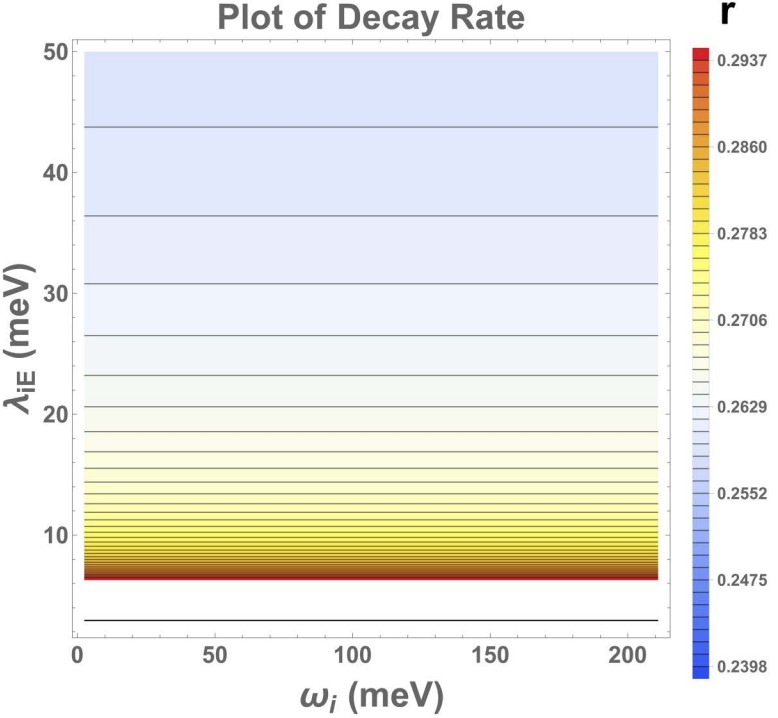

**Fig 9. Shows a contour plot illustrating how environmental structure, reorganization energy, and vibrational modes influence decay rates.**
In the red region, where reorganization energy is weak, decay rates are higher, driven narrow vibrational modes. As reorganization energy approaches thermal energy (yellow region), decay rates remain stable with denser contour lines, reflecting nuanced dynamics. In the blue region, where reorganization energy exceeds thermal energy, decay rates decrease, contour spacing increases, and vibrational modes play a diminished role compared to the energy difference between the donor and acceptor. The plot highlights the transition from weak to strong coupling regimes and their impact on transfer dynamics.

interplay of quantum states. This comprehensive analysis not only broadens our understanding of electron tunneling phenomena but also accentuates the crucial role of vibrionic modes and structured environments in modulating and enhancing quantum interactions within biological complexes.

## Conclusion

In this investigation, we utilized the NMQSD methodology to explore the dynamics of SARS-CoV-2 infection, specifically examining the role of vibration-assisted electron tunneling within structured environments. By implementing the Non-Markovian Stochastic Schrödinger Equation, our study delved into the microscopic mechanisms that underlie viral infection, focusing on how electron tunneling, influenced by various biological parameters, facilitates the interaction between the virus's spike protein and the ACE2 receptor.

Our findings highlight the role of non-Markovian dynamics in accurately capturing the electron tunneling processes critical to viral infections within structured environments. Tunneling probabilities are minimal at lower coupling strengths but increase substantially as the coupling between donor and acceptor levels intensifies. Vibrationally assisted electron tunneling is observed to be most effective due to the vibrational modes of the virus in the intermediate coupling regime. In contrast, under very strong coupling conditions, tunneling becomes coherently driven. The structured environment sustains positive coherence even at lower coupling strengths and minimal vibrational modes. Extending beyond the Marcus-Jortner rate theory, our study reveals that tunneling remains intact even in regimes surpassing its predictive limits.

However, under very strong environmental coupling, the influence of the environment significantly reduces electron tunneling rates.

Finally, the results underscore the critical role of vibrational modes in influencing quantum dynamics. Specifically, vibrational modes that are comparable to the energy differences between donor and acceptor sites significantly enhance both population and coherence dynamics. In contrast, off-resonance vibrational modes tend to suppress tunneling dynamics, thereby shaping the quantum mechanical processes involved in viral entry into host cells.

## Supporting information

**S1 Fig. Donor population with and without vibrational coupling (mean 95% CI).** Note: All statistical analyses are presented in the Supporting Information (SI).
(PNG)

**S2 Fig. Shows Sensitivity of vibronic enhancement across electronic coupling and vibrational frequency.**
(PDF)

**S3 Fig. Time-averaged vibronic enhancement across electronic coupling and frequency.**
(PDF)

**S4 Fig. Vibronic sensitivity of the time-averaged rates.**
(PDF)

## Author contributions

**Conceptualization:** Muhammad Waqas Haseeb, Mohamad Toutounji.

**Data curation:** Muhammad Waqas Haseeb.

**Formal analysis:** Muhammad Waqas Haseeb, Mohamad Toutounji.

**Funding acquisition:** Mohamad Toutounji.

**Investigation:** Muhammad Waqas Haseeb, Mohamad Toutounji.

**Methodology:** Muhammad Waqas Haseeb.

**Project administration:** Mohamad Toutounji.

**Resources:** Mohamad Toutounji.

**Supervision:** Mohamad Toutounji.

**Validation:** Muhammad Waqas Haseeb, Mohamad Toutounji.

**Writing – original draft:** Muhammad Waqas Haseeb.

**Writing – review & editing:** Muhammad Waqas Haseeb, Mohamad Toutounji.

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
