## [Decision Letter · Decision Letter 0]

5 Oct 2025

Dear Dr. Toutounji,

Please submit your revised manuscript by Nov 19 2025 11:59PM. If you will need more time than this to complete your revisions, please reply to this message or contact the journal office at plosone@plos.org. Please include the following items when submitting your revised manuscript:

We look forward to receiving your revised manuscript.

Kind regards,

Dr. Dola Sundeep

Academic Editor

PLOS ONE

Journal Requirements:

https://journals.plos.org/plosone/s/file?id=ba62/PLOSOne_formatting_sample_title_authors_affiliations.pdf....

“This work was funded by UAE University Research Affairs under grant number G-00003550.”

4. In the online submission form you indicate that your data is not available for proprietary reasons and have provided a contact point for accessing this data. Please note that your current contact point is a co-author on this manuscript. According to our Data Policy, the contact point must not be an author on the manuscript and must be an institutional contact, ideally not an individual. Please revise your data statement to a non-author institutional point of contact, such as a data access or ethics committee, and send this to us via return email. Please also include contact information for the third party organization, and please include the full citation of where the data can be found.

6. Please update your submission to use the PLOS LaTeX template. The template and more information on our requirements for LaTeX submissions can be found at http://journals.plos.org/plosone/s/latex....

7. Thank you for stating the following in the Acknowledgments Section of your manuscript:

“This work was funded by UAE University Research Affairs under grant number G-00003550.”

“This work was funded by UAE University Research Affairs under grant number G-00003550.”

Additional Editor Comments (if provided):

Dear Dr. Mohamad Toutounji,

Manuscript Number: PONE-D-25-45999

Title: Non-Markovian Electron Tunneling in SARS-CoV-2 Virus Infection in Structured Environments

Thank you for submitting your manuscript to PLOS ONE. The reviewers have now provided their evaluations. After considering their comments, the Editorial Board decision is Major Revisions.

The reviewers have identified significant issues related to theoretical justification, data interpretation, and clarity of exposition that must be addressed before further consideration for publication. We therefore invite you to submit a substantially revised version of your manuscript. Please ensure that you provide a detailed, point-by-point response to each reviewer comment in your revision.

We believe that, with these revisions, your work could make a valuable contribution to the field.

Thank you for choosing PLOS ONE for your research. We look forward to receiving your revised submission.

Sincerely,

Dr. Dola Sundeep

Academic Editor

PLOS ONE

Reviewers' comments:

Reviewer's Responses to Questions

**Comments to the Author**

1. Is the manuscript technically sound, and do the data support the conclusions?

Reviewer #1: Yes

Reviewer #2: No

Reviewer #3: Yes

2. Has the statistical analysis been performed appropriately and rigorously?

Reviewer #1: Yes

Reviewer #2: No

Reviewer #3: I Don't Know

3. Have the authors made all data underlying the findings in their manuscript fully available?

Reviewer #1: Yes

Reviewer #2: No

Reviewer #3: Yes

4. Is the manuscript presented in an intelligible fashion and written in standard English?

Reviewer #1: Yes

Reviewer #2: No

Reviewer #3: Yes

Reviewer #1: Dear author,

The article titled "Non-Markovian Electron Tunneling in SARS-CoV-2 virus infection in

Structured Environments" was reviewed and please correct the following.

References 1-3, 10-17, 20, 21, 25, 28, 29, 31-33, 35, 36, are old, please use new references.

Kind regards

Reviewer #2: The manuscript addresses an interesting and novel question but requires major revision. The work is not yet technically sound, lacks statistical rigor, does not comply with PLOS data availability requirements, and contains presentation issues.

1. Technical Soundness and Data Support

Line 110–150 (Methods, Eq. 3–11): Numerical implementation details are missing. Please specify the integrator used, time step, ensemble size (number of trajectories), and convergence checks. Without this, the reproducibility of results is unclear.

Line 250–300 (Figure 3 description): Population transfer is shown as color plots, but no error bars or variance measures are provided. Please include uncertainty quantification.

Line 370–400 (Results, Figure 5):The phrasing “particularly in with the virus” is grammatically incorrect and unclear; revise to “particularly with the virus.”

Line 420–460 (Discussion of Figures 7–8): The role of resonance and off-resonance vibrational modes is asserted, but no sensitivity tests are shown. Please demonstrate robustness to small frequency shifts.

2. Statistical Analysis

Line 90–100 (Methods, ensemble averages): The manuscript notes “ensemble-averaged density matrix” but does not report the number of trajectories (N) used. Please add this information and provide an error analysis.

Line 300–320 (Eq. 28–29, definition of ∆P and κ): These observables should be reported with confidence intervals or error bars, since they are derived from stochastic simulations.

3. Data Availability

Line 1240–1260 (Data Availability Statement): The current text states, “data available upon justified request.” This does not comply with PLOS ONE policy. Please deposit all underlying data (raw simulation outputs, processed data behind figures, and ideally code) in a public repository (e.g., Zenodo or GitHub with DOI) and update the statement accordingly.

4. Presentation and English

Line 220 (Figure 1 caption): Check for the spelling mistake “protien” instead of “protein.”

Line 610 (Figure 5 caption): “both case exhibit” instead of “both cases exhibit.”

Line 790 (Figure 8 caption): “negative cohereces” instead of “negative coherences.”

Line 850 (Discussion): “Futhermore” instead of “Furthermore.”

Throughout Results and Discussion:Sentences are often long and complex (e.g., Line \~270–290, spectral density discussion). Please simplify for clarity.

Notation: Units are inconsistently reported (cm⁻¹ vs eV). Please provide conversions consistently and define all symbols (γ, γi, γiE) on first use.

Reviewer #3: Kindly find comments to editor below

.

Reviewer #1: No

Reviewer #2: No

Reviewer #3: No

---

## [Author Response · Author response to Decision Letter 1]

26 Nov 2025

We have addressed all the comments of reviewer # 2. We have also created is additional Supporting information to present all our statistical analyses. We also updated the data availability statement.

---

## [Decision Letter · Decision Letter 1]

26 Dec 2025

Dear Dr.  Toutounji,

Thank you for submitting your manuscript to PLOS ONE. After careful consideration, we feel that it has merit but does not fully meet PLOS ONE’s publication criteria as it currently stands. Therefore, we invite you to submit a revised version of the manuscript that addresses the points raised during the review process.

We look forward to receiving your revised manuscript.

Kind regards,

Dola Sundeep

Academic Editor

PLOS One

Journal Requirements:

1. If the reviewer comments include a recommendation to cite specific previously published works, please review and evaluate these publications to determine whether they are relevant and should be cited. There is no  equirement to cite these works unless the editor has indicated otherwise.

Additional Editor Comments:

Dear Mohamad Toutounji,

We have now received the required number of reviewer reports for your revised manuscript:

PONE-D-25-45999R1

“Non-Markovian Electron Tunneling in SARS-CoV-2 Virus Infection in Structured Environments.”

After careful consideration of the reviewers’ comments, the Editorial Board has reached a decision of Minor Revision.

Please address the remaining reviewer comments and provide a concise, point-by-point response detailing the changes made in your manuscript. All revisions should be clearly indicated in the revised version.

We look forward to receiving your updated submission.

Sincerely,

Dr. Dola Sundeep

Academic Editor

Reviewer's Responses to Questions

**Comments to the Author**

Reviewer #1: (No Response)

Reviewer #2: (No Response)

2. Is the manuscript technically sound, and do the data support the conclusions?

Reviewer #1: Yes

Reviewer #2: (No Response)

3. Has the statistical analysis been performed appropriately and rigorously?

Reviewer #1: Yes

Reviewer #2: (No Response)

4. Have the authors made all data underlying the findings in their manuscript fully available?

Reviewer #1: Yes

Reviewer #2: (No Response)

5. Is the manuscript presented in an intelligible fashion and written in standard English?

Reviewer #1: Yes

Reviewer #2: (No Response)

Reviewer #1: Dear Author,

The article titled "Non-Markovian Electron Tunneling in SARS-CoV-2 Virus Infection in Structured

Environments" has been reviewed.

Please make the following correction:

References 3, 13, 26, and 29 are old, please use new references.

Kind regards

Reviewer #2: (No Response)

.

Reviewer #1: No

Reviewer #2: No

---

## [Author Response · Author response to Decision Letter 2]

1 Jan 2026

We thank the reviewer for this valuable suggestion. We have systematically updated the bibliography throughout the manuscript. All items flagged as “old” (Refs.~ 3, 13, 26, 29, ) were reviewed and, where appropriate, replaced with recent peer-reviewed sources from the last few years.

All updates are visible in the revised reference list and highlighted in orange color in the tracked-changes version.

---

## [Decision Letter · Decision Letter 2]

22 Feb 2026

Non-Markovian Electron Tunneling  in SARS-CoV-2 Virus Infection in Structured Environments

PONE-D-25-45999R2

Dear Dr. Toutounji,

We’re pleased to inform you that your manuscript has been judged scientifically suitable for publication and will be formally accepted for publication once it meets all outstanding technical requirements.

Reviewer 2 has sugested a new list of references to be checked and perhaps updated (see below). Please consider these suggestions as optional.

Dear author,

The article titled "Non-Markovian Electron Tunneling in SARS-CoV-2 Virus Infection in Structured Environments" was reviewed. This article provides useful information for its readers. Please make the following corrections:

References 2, 7, 16, 19, 24, 27, 30, and 38 are old, please use new references.

Kind regards

Kind regards,

Dennis Salahub

Academic Editor

PLOS One

Additional Editor Comments (optional):

Reviewers' comments:

Reviewer's Responses to Questions

**Comments to the Author**

Reviewer #1: (No Response)

Reviewer #2: All comments have been addressed

2. Is the manuscript technically sound, and do the data support the conclusions?

Reviewer #1: Yes

Reviewer #2: (No Response)

3. Has the statistical analysis been performed appropriately and rigorously?

Reviewer #1: Yes

Reviewer #2: (No Response)

4. Have the authors made all data underlying the findings in their manuscript fully available?

Reviewer #1: Yes

Reviewer #2: (No Response)

5. Is the manuscript presented in an intelligible fashion and written in standard English?

Reviewer #1: Yes

Reviewer #2: (No Response)

Reviewer #1: Dear author,

The article titled "Non-Markovian Electron Tunneling in SARS-CoV-2 Virus Infection in Structured Environments" was reviewed. This article provides useful information for its readers. Please make the following corrections:

References 2, 7, 16, 19, 24, 27, 30, and 38 are old, please use new references.

Kind regards

Reviewer #2: (No Response)

.

Reviewer #1: No

Reviewer #2: No

---

## [Editor Report · Acceptance letter]

PONE-D-25-45999R2

PLOS One

Dear Dr. Toutounji,

I'm pleased to inform you that your manuscript has been deemed suitable for publication in PLOS One. Congratulations! Your manuscript is now being handed over to our production team.

Kind regards,

on behalf of

Dr. Dennis Salahub

Academic Editor

PLOS One